# Winning Conditions for the Adoption and Maintenance of Long-Term Healthy Lifestyles According to Physical Activity Students

**DOI:** 10.3390/ijerph182111057

**Published:** 2021-10-21

**Authors:** Laurence Couture-Wilhelmy, Philippe Chaubet, Tegwen Gadais

**Affiliations:** 1Département des Sciences de l’activité Physique, Université du Québec à Montréal, Montréal, QC H2X 1Y4, Canada; Couture-Wilhelmy.laurence@courrier.uqam.ca (L.C.-W.); chaubet.philippe@uqam.ca (P.C.); 2Chaire UNESCO en Développement Curriculaire, Université du Québec à Montréal, Montréal, QC H2X 1Y4, Canada; 3Centre de Recherche Interdisciplinaire sur la Formation et la Profession Enseignante (CRIFPE), Université du Québec à Montréal, Montréal, QC H2X 1Y4, Canada

**Keywords:** healthy lifestyles, health education, physical activity, qualitative approach, long term, maintenance

## Abstract

The literature has not yet well documented the relative elements of the adoption of healthy lifestyle habits (HLHs) over the long term. More especially, researchers are calling to complete the corpus with qualitative or mixed estimates that would allow them to better explain the conditions necessary for the adoption or maintenance of HLHs over the long term. The present study seeks to understand the winning conditions for the adoption and maintenance of HLHs. Semi-structured group interviews were conducted with three groups of university students (two in Bachelor’s degree in physical education (PE) and one in Master’s degree in physical activity science), all in favor with HLHs. The results identify some dominant winning conditions in the adoption and maintenance of HLHs, such as the role of the family environment, the role of diversity and choice in physical activities during youth, the physical and social environment, autonomy and also mental health, which is closely linked with physical health. Results were modelled in the form of an ideal pathway, which traces the impact of winning conditions from childhood to adulthood. The originality of this study stands out, among other things, because of its innovative methodology; therefore, this study opens the door to future qualitative research in the field. Investigating pathways, considering the different phases of development of children and adolescents to identify factors of change and maintenance of HLHs now seems to be an interesting and necessary avenue for research in the field.

## 1. Introduction

This study focuses on the winning conditions for the adoption and maintenance of healthy lifestyle habits (HLHs) by university students in training in the field of physical activity (physical education and kinesiology). The project uses a qualitative, backward-looking methodology to document the elements and mechanisms of their life experience that students retrospectively perceive as important in their awareness of HLHs and in the adoption and maintenance of some of them. The interest in the issue of maintenance stems from the lack of evidence in health promotion about what works in the long term to build HLHs. Therefore, it is interesting to understand what types of experience play a role in the winning conditions during the course of life. For example, are they cumulative, repeated, one-time but transformative experiences?

By “winning conditions”, we refer to all kinds of factors that have a positive impact on the HLHs in the participants’ journey. The exploratory context of this study leads us not to restrict ourselves to one category of factors, but rather to be inclusive in all their variety, with the greatest possible openness. Thus, following the example of the life course theory [1], our approach seeks to “grasp the logics that structure diverse trajectories” [1] (p. 34) and to “understand the interactions that link them to one another” [1] (p. 34). This is conducted with the knowledge that individual trajectories are fragmented by moments of stability, improvement and decline in relation to HLHs [2].

### 1.1. Improving Health by Promoting Healthy Lifestyles

The growing interest in health promotion has its origins in the fight against numerous chronic diseases and the increased sedentary lifestyle of the population, because children are also more likely to remain obese as adults [3,4,5,6]. In Canada, only one-third of Canadian youth meet the recommended PAP guidelines of 60 min of moderate-to-high intensity daily physical activity participation (PAP) [3,7]. In response, several proposed interventions have been implemented to increase youths’ PAP to produce immediate or longer-term health effects [8]. In fact, although HLHs have been well documented for decades, we still do not know enough about which factors should be targeted for the promotion of health in our populations [3,9,10]. Indeed, the trend indicates that young people are living increasingly sedentary lifestyles and are less and less invested in regular PAP [11].

### 1.2. Current State of Research on Healthy Lifestyles and Theoretical Models in Health Promotion

The literature offers several definitions and components of HLHs that differ according to contexts and study objectives. Thus, authors often target determinants of health that they deem relevant to their research [12]. Consequently, it is possible to identify several theoretical models for understanding HLHs and the development of winning conditions for their adoption and maintenance. Some are predictive and others are prescriptive or are part of an ecosystem principle. First, some so-called “predictive” theories attempt to identify and explain the determinants of the adoption of a health-related behavior, for example Azjen’s theory of planned behavior [13] or Godin’s integrative model [14]. Ajzen’s [13] theoretical model focuses on the notion of intention to adopt a behavior. This behavioral intention is influenced by various factors and must first and foremost be planned and decided upon in order for change to occur. The higher the degree of strength of the intention, the more positively efforts are turned towards the commitment of the behavior, the more possible its adoption becomes [13]. In a complementary manner, Godin [15] proposes a model of a holistic view of health change. In his integrative model, similarly to Ajzen’s, intention is key. Behavioral change can occur through the perception of behavioral control when adoption does not come from the individual’s will [14]. Godin goes further with the addition of external variables [15] that include individual characteristics, such as age, gender and personality, as well as environmental characteristics (physical and social), such as culture, socioeconomic background, etc. Ultimately, Azjen’s and Godin’s theories help us understand the phenomena of intentionality related to the adoption and maintenance of HLHs, while incorporating the individual and environmental aspect of the person. Second, so-called “prescriptive” theories attempt to understand the adoption and maintenance of HLHs by promoting the implementation of a behavioral change process. Deci and Ryan’s [16] theory of self-determination proposes, for example, a continuum of motivation, ranging from amotivation (lack of motivation), through the full spectrum of extrinsic motivation, to intrinsic, or self-determined motivation [17]. One of the key propositions in this theory is that self-determined motivation satisfies three basic psychological needs in the individual, namely, autonomy, a sense of competence and social belonging, which would increase the chances of maintaining HLHs. Third, the social-ecological model attempts to understand the interrelationships between the individual and their various ecosystems. Bronfenbrenner developed the social ecological model [18], which was later taken up by Sallis and Owen [19] in HLHs and health promotion. First, this model has several levels that influence health: intrapersonal, interpersonal, organizational, community and political. Second, sociocultural factors and physical environments can influence HLHs on more than one level. Furthermore, there appears to be an interaction between these different levels of influence; the different variables work together and not in isolation.

From these types of models, it is possible to differentiate several trends in the theoretical models of health promotion and HLHs. For example, it is possible to clarify the reasons for an intention to adopt a HLH by identifying the factors underlying this intentionality, such as perceived control (self-efficacy), perceived norms (peer and family influence) and attitudes (positive attitude towards the behavior) [14]. In addition, interesting links can be made with the role of peer-guided motivation, feelings of self-efficacy and autonomy, as in the case of self-determination theory [16]. Finally, the social ecological model [19] comes to integrate everything in an ecosystemic way. These theories can also be differentiated by their type of intervention and by their targets. In other words, there are theories that are more person-centered, such as Deci and Ryan’s and Ajzen’s. On the other hand, we have the socio-ecological model of Sallis and Owen, which places the environmental dimension as a priority. Between the two, we have Godin’s integrative model, which unifies these two axes in their vision of an efficient intervention.

However, none of these models can fully and satisfactorily explain why an individual adopts or maintains HLHs over time. Consequently, it is illusory to think that a single model could satisfy the requirements and perfectly guide future interventions [14,20]. Added to this are the numerous limitations (e.g., lack of long-term evidence, need to complete research with mixed or qualitative designs) that have been raised in the various health promotion studies for several decades [21,22,23,24] that lead us to question research methods, as no model seems to be able to explain how HLHs are adopted and, especially, maintained [25]. One of the alternatives proposed by researchers [22,26,27] that have been little explored by the literature most likely concerns the type of methodology employed.

### 1.3. Limitations of Studies and Their Methodologies to Study Long-Term HLHs

Despite health promotion efforts, authors note a lack of sustainability and maintenance in health promotion intervention outcomes [21,22,23,24]. Current programs, in most cases, result in short-term improvements that tend to fade within a few weeks after the intervention ends [8]. Spinola and Castro’s [23] study supports this by demonstrating a lack of long-term evidence. In addition to these findings, few results address the best strategies to adopt when it comes to maintaining HLHs. Thus, the evidence for the effectiveness of health promotion strategies is weak, both in terms of implementation, policy and practice [24].

The development of a HLH and its maintenance occurs in a process that may take time to settle, if only because it may involve “relapses” to old behaviors [28]. In this sense, current methodologies do not allow for relevant findings, as they do not seem to be adapted to the journey of a young person, who goes through several developmental stages and a multitude of conditions that shape his or her HLH choices [29].

Furthermore, there is a need to supplement numerous quantitative studies with qualitative studies to find precisions about the phenomena and explore perspectives of HLHs. Indeed, many quantitative studies have been conducted to address the poor lifestyle habits of youths [22,27] and, generally, distinguish between experimental, cross-sectional studies, randomized controlled trials, in particular [30]; many provide data on the effectiveness of individual programs [25]. However, these studies have important limitations, as mentioned by these authors. On the one hand, they mostly focus on interventions that are very localized in space and limited to a specific intervention period. On the other hand, their essentially quantitative analyses make generalizations that are not very valid, since they rarely take into account the fact that participants are often influenced by the characteristics of the environment set up in the intervention [31]. These research designs focus on numerical (quantitative) observables, without delving qualitatively and concretely into what really matters to individuals. In particular, how to maintain a HLH engaged or enhanced by these specific time-limited interventions [9].

### 1.4. Relevance of the Study

As discussed above, current studies of HLH adoption and maintenance have methodological limitations, including a lack of long-term evidence, due to designs that are often limited to the intervention period, and predominantly quantitative approaches that do not allow researcher to attain a fine-grained understanding of the phenomena. Here, then, there are three methodological considerations in an attempt to address these limitations.

#### 1.4.1. Documenting the Positive Experiences

It is relevant to document “what seems to work” for the adoption and maintenance of HLHs, i.e., what promotes a HLH rather than what hinders it, because it is not said that it would be enough to “reverse” the conditions that hinder it so that, all of a sudden, people adopt and maintain VHS. The approach of exploring the successes and the positive that exist in the individual has a solid scientific basis [32], which is found in the method of appreciative inquiry, for example, looking for the positive—what works and not what harms [33]. Thus, appreciative inquiry is in line with positive psychology on the idea of highlighting strengths, which are more motivating than weaknesses and more conducive to change in human beings. Cooperrider and Whitney were influenced by this positive approach in research in the fields of sports, medicine, behavioral sciences, etc., proving that positive images have a considerable impact in human psychology [34].

#### 1.4.2. Use of a Long-Term Methodology through a Qualitative Design

The idea is to overcome the temporal limitation of the short term, which fails to capture the fluctuations in adoption, dropouts, resumption, or relapse that are so important to understanding the strength and dynamics of maintenance over time. This type of approach also avoids deploying complex, time-consuming and costly methodological follow-ups. Consequently, it seems relevant to study HLHs through a qualitative approach and from a long-term perspective, which has never been performed, to our knowledge. A better understanding of what promotes the maintenance of HLHs could help inform future program or policy decisions [25].

#### 1.4.3. Use of a Methodology That Works with Reverse Engineering (Chaubet et al., 2016)

In this study, the objective is not to follow people over several years, as in a longitudinal study, but rather to reconstitute the major inflections of the subjects’ experience backwards, starting from the account they are able to give, accompanied by a researcher. The analogy underlies the method of industrial or computer reverse engineering, that is, taking a finished and functional product, then dismantling it piece by piece to understand its operating principles. Thus, the study wishes to highlight triggering situations that have challenged the subjects and may have contributed to changes in or maintenance in people’s habits. The whole is inspired by the reverse engineering approach of the facilitating conditions of certain human phenomena proposed by Chaubet et al. [35].

### 1.5. Research Question and Objectives

Therefore, this article aims to answer the following question: What are the winning conditions for adopting and maintaining healthy lifestyle habits according to the stories told and illustrated by students in physical and health education and in physical activity sciences?

We identified three objectives to answer this question:To describe the pathways that led physical and health education (PE) and Master of Science students in Physical Activity (M) to adopt and maintain HLHs and to characterize the forms of change in or maintenance of these HLHs;To identify the winning conditions in the adoption and maintenance of HLHs among these students and characterize them;To integrate the winning conditions into a conceptual model of HLH adoption and maintenance.

## 2. Methods

### 2.1. Research Design

This research project is based on a qualitative–interpretive methodology, that is, the elements that emerge from the data collected are derived from the participants’ understanding of the phenomenon under study, according to their experience. This inductive framework then becomes a scientific tool to shed light on the determinants of the adoption and maintenance of HLHs. It is a qualitative process inscribed in a dynamic that seeks to make sense of and understand the complexity of a phenomenon, while remaining rooted in the dialectic of interpretations and representations made by the participants [36]. The human experience, whether real or imagined, is related to a statement, which makes it possible to seek an understanding of the phenomenon [37].

### 2.2. Participants and Recruitment

The intended participants were identified in a convenience sample, based on intentionality criteria [38]. Originally, four group profiles were selected: Bachelor’s degrees in PE education, kinesiology, dance education and Master’s degrees in physical activity science. The COVID-19 pandemic restricted access to individuals. For feasibility reasons, the analysis was reduced to the three groups already interviewed: two in PE (N-PE1 = 5 and N-PE2 = 7) and one in MSc (N-M = 13).

Faculty members responsible for the university courses in the programs mentioned facilitated access to their group courses. The student participants share two characteristics: (1) their disciplines challenge them on issues of HLHs by requiring them to consider the physical and mental health of the individuals with whom and for whom they are training; (2) they also have an experience of personal reflection on these issues, with their program providing additional tools and opportunities for reflection. Therefore, this sample is purposive, accessible and relevant to the research purpose and question, a critical aspect for the scientific validity of a study [39].

### 2.3. Instrumentation and Data Collection

The main instrument used for this research was the semi-structured group interview, audio recorded and verbatim transcribed. A total of three interviews lasting approximately 40 min were conducted two in PE and one in M. The semi-structured interview method was chosen in order to allow participants to have the freedom to delve into their journeys, with their own set of stories and according to their sensitivity to the field. An interview guide helped to keep the exchange within the following three themes: the participants’ experiences with HLHs (their practice); behavioral changes (before and after); the facilitating elements in their journey to adopt them (what made the difference, what helped foster change) (see Appendix F).

### 2.4. Analyses

The data analysis was conducted in three stages. A first level of analysis isolated concrete experiences experienced by and exemplified by the individuals, indicating changes in participants’ behavior. From these “islands of change”, the analysis expanded its focus to the sources and/or conditions surrounding these changes [35]. Thus, an initial code grid was formed, a mixture of pre-constructed categories (change and surrounding conditions of those changes) and self-imposed categories related to the goals (form of harmful lifestyle habits, signs of maintaining HLHs, signs of not maintaining, winning condition of HLHs, etc.).

The second stage was part of a completely inductive movement, consisting of a more detailed understanding of each of the phenomena identified. This inductive work, by conceptualizing categories [37], strongly inspired by the progressive implementation of a “grounded theorization”, but without successive returns to the field [40], produced, little by little, a modeling of several phenomena relevant to the study.

Finally, in the third and last step, recurring elements were noted and links were established between the different accounts of the participants (which we also call “stories”, because that is often the form they take), at different moments of their experience. From there, a “linking” exercise identified similarities, dependencies and hierarchies among the elements of analysis, in order to create an organized and structured model [40] that would allow us to establish typologies and map an overall picture of the results.

## 3. Results

This section presents the results of the study by following our research objectives. Our first research objective—to describe the adoption and maintenance pathways of HLHs—was addressed during the analysis stage by isolating the concrete experiences of participants using pre-constructed categories (e.g., forms of HLHs, signs of maintenance, winning conditions of HLHs and changes).

The following results correspond to the second research objective, which was to identify the types of winning conditions for the adoption and maintenance of HLHs. To achieve this, the analytical construction was carried out in three steps: (1) identify these winning conditions and their predominance for each of the profiles, in the form of key characteristics (Appendix A and Appendix B); (2) create a schematic representation with these same characteristics, but, this time, establish influences between them (Figure 1, Figure 2, Figure 3 and Figure 4); (3) synthesize these results in a conceptual map and a preliminary modeling that includes the two profiles under study (Figure 5 and Figure 6).

### 3.1. Summary of the Winning Conditions for the Adoption and Maintenance of Healthy Lifestyle Habits

Appendix A and Appendix B summarize the dominant characteristics of our participants’ adoption and maintenance of HLHs, merging the two profiles PE and M1. It partially addresses our second research objective—to identify and characterize the winning conditions for HLH adoption and maintenance. A table for each profile was initially created (Appendix A and Appendix B) to arrive at this overall representation (Figure 5). The main characteristics of the two profiles that emerged during the interviews are identified and illustrated by blocks of different colors, which correspond to decreasing levels of recurrence according to the number of participants (blue, green, orange, yellow and purple).

There are five levels of key characteristics. The first includes the two most important key characteristics of participants’ adoption and maintenance of HLHs, the relationship between physical and mental health and the family environment. The second represents the social environment. The third includes the link among enjoyment, motivation and self-efficacy, as well as the integration of the outdoors into one’s life. The fourth level consists of the intrinsic awareness of the benefits of HLHs. Finally, the fifth level has three characteristics: the physical environment, the balance between the different spheres of our lives and the routine (categories are defined in Appendix C).

### 3.2. Diagram of the Winning Conditions for the Adoption and Maintenance of Healthy Lifestyle Habits

Subsequently, a schematic representation of the key characteristics was developed based on the influences they have on each other to better understand the role of these winning conditions. This representation helps to address our second research objective—to identify the types of winning conditions for HLH adoption and maintenance. As with Appendix A and Appendix B, a diagram for both profiles (Appendix D and Appendix E) was initially created to arrive at this synthesis. To facilitate understanding of the synthesis and its concepts, we propose, in the next sections, a gradual schematic and conceptual construction.

#### 3.2.1. Blue Winning Conditions (Very Strong) and Their Links

##### A Strong Link between Physical Activity Participation and Psychological Health

There are three major elements in this schematic construction (Figure 1). First, PAP and psychological health are at the very heart of the analysis, as they are the elements most exposed by our participants. A priori, they are individual dimensions of health; therefore, they are sought-after goals, in terms of HLHs. The results show that they are also winning conditions for the adoption and maintenance of HLHs, because they act in a bidirectional and cyclical way towards each other, as indicated by the arrows in the figure.


*YÉD2: I have always found over the years that I felt better and had better overall mental health when I exercised just a little bit, every day. This is something I discovered quite early on, but I confirm it [...]*



*MAK1: Sport has always been part of my life since I was little. So it’s something I do for fun, it’s a hobby, it’s a need to feel good. So it’s been integrated since I was really little.*


##### The Family with Its Support, Diversity and Role Model as a Predominant Element in the Adoption of Healthy Lifestyle Habits

The third major element in this pattern is the blue circle of family, which implies support, stability and an inspirational model. One explanation for the connection among these three elements is a family environment that has been able to provide diversity for their child.


*SÉD1: [...] Every weekend we did something new, especially in nature. We’d go hiking, we’d go mushroom picking, unusual things, fishing, hunting. Uh they signed me up for field hockey, soccer, baseball when I went to the States. Uh... I had a chance to try out a lot of sports [...]*


This family environment also provides ongoing support.


*ÉD2: [...] Of course I couldn’t do all the sports at the same time, I would have liked that, but they said, Perfect, we’ll support you. And I did other sports as well and every time they were behind me.*


#### 3.2.2. Green (Strong) Winning Condition and Its Links

##### The Social Environment as a Lever for the Physical Activity Participation

The second level of recurrence is the social environment (Figure 2). It has a bidirectional link to the PAP. Some mentioned that peers pushed them to move while others invested in PAP because of their social aspect. This winning condition for adopting and maintaining HLHs also has an important link to motivation.


*OÉD2: [...]… sometimes I was not motivated. Then my friends would tell me “oh, let’s go, let’s go do some sports” and I would go and finally I was happy to go but otherwise I wouldn’t have gone by myself.*


The physical environment and leisure services are also winning conditions (in purple).


*NK1: And when you go outside, well, you find a lot of... I played baseball very quickly from the age of 5–6 at the local park and that made me want to play [...] Without the social infrastructure where people were outside, there might not have been as many hooks either.*



*Mk1: [...] Everyone is there at the same time: the arena, the baseball field, the park. So the kind of location creates a kind of excitement to go outside and play with our friends outside and then bike to our parks.*


#### 3.2.3. Orange (Medium Strength) Winning Conditions and Their Links

##### Pleasure, Motivation and Self-Efficacy Intimately Linked

The factors rated as moderately strong are then enjoyment, sense of self-efficacy, motivation and peers (Figure 3). Three of these interact. For example, a sense of competence can lead to increased enjoyment of the PAP and thus motivation.


*AÉD2: It was also a need to... Not a need but actually I felt successful, I felt competent in what I was doing so I was just enjoying doing the sport by myself.*


##### Integration of the Outdoors

Another winning condition judged “moderately strong” is the place of the outdoors in the lives of participants (Figure 3). It is influenced by the pleasure experienced when they integrate it into their lives and it refers as much to sports practiced outdoors as to outdoor activities such as camping. The outdoors also has links to PAL and psychological health.


*MTK1: But what I found important to integrate is the outdoors, so to really have a contact with nature, that’s primordial and also mental health. I think that stress is such a major problem.*


#### 3.2.4. Yellow Winning Condition (Fairly Strong) and Its Links

##### Intrinsic Awareness of the Benefits of Physical Activity on Physical and Psychological Health

The dominant element for this level is the intrinsic awareness of the benefits of HLHs (Figure 4). It refers to the individual’s awareness of their choices in terms of HLHs, that is, they realize the importance of their actions. The word intrinsic is used because, according to the participants, it is an awareness that was once influenced by external elements, such as family, and, now, comes from within.


*VIK1: It was just implanted, it was just natural, I didn’t really ask myself any questions. [...] And when I got to college, university, I just realized how important it was.*


The link between intrinsic awareness and mental and physical health is also present:


*AK1: And at the university, with kin[esiology] it was not really my appearance, but it was more when I train, I have more energy, as I feel better, I have less headache. So it seems that my evolution was more external and now it seems more internal.*


#### 3.2.5. The Purple Winning Conditions (Strong Enough) and Their Links

##### Role of Routine, Balance (Physical-Psychological-Social)

Finally, factors strong enough to still be cited by participants were routine and balance (Figure 5). Analyses show that routine plays a role in achieving regular and/or daily PAP. As for balance, we see a direct link to psychological health and participants refer to a lifestyle that includes time for PAP, psychological health and social life.


*GÉD1: [...] I changed that very habit. So I opened a little more time for my family, for my friends and to go towards new activities. And that, in my eyes, is a healthier way of life than what I had before, where I was focused on one thing and one thing only: sports. So, this is a turning point. [...] But I still keep in mind the learning that I did, so I always keep time for the other spheres of life.*


#### 3.2.6. Summary of the Winning Conditions for the Adoption and Maintenance of Healthy Lifestyle Habits

Figure 5 illustrates the complete synthesis of the winning conditions for the adoption of HLH maintenance found among participants in both profiles. The different colors and thickness of the circle express the strength according to recurrence (for ease of understanding, these are the same colors as in Appendix A and Appendix B). The solid black arrows show which characteristics may have acted on another. Dotted black arrows express a bidirectional link between two or more characteristics. Finally, quotes from participants directly illuminate the type of link between concepts on the arrows.

### 3.3. Summary of Findings

Appendix A and Appendix B and Figure 5 summarize the winning conditions for the adoption and maintenance of HLHs among the two participant profiles. Overall, the predominance of PAP belt functioning and mental health can be seen. These two dimensions of health are influenced by many winning conditions, such as family, social and physical environment, and the triangular link between pleasure, motivation and self-efficacy. Furthermore, intrinsic awareness of the benefits of a HLH and the integration of the outdoors have an impact on both PAP and psychological health. Finally, having a routine implemented in one’s schedule seems to be a winning condition for maintaining a PAP level; similarly, the balance between the spheres of one’s life helps the participants’ mental health.

## 4. Discussion

In this qualitative study, participants identified numerous factors that had an impact on the adoption and maintenance of their HLH that is different from and similar to numerous quantitative studies. After describing the pathways and then characterizing the forms of change that led students from two profiles (PE and M) to adopt and maintain HLHs, the winning conditions for adoption and maintenance were identified for these participants (Appendix A and Appendix B) by showing these key concepts and their influential relationships (Figure 5). The third objective was to integrate these winning conditions into a conceptual model of HLH adoption and maintenance (Figure 6). To achieve this, a conceptualization of these mechanisms was made, according to the paths of the two profiles under study. It is organized to understand their evolution and degree of importance. This modelling becomes relevant in order to shed light on the determinants on which to act and the appropriate periods to act, in order to help the adoption and maintenance of HLHs.

### 4.1. Preliminary Model of Healthy Lifestyle Habit Adoption and Maintenance

This model (Figure 6) can be understood according to four pillars of HLH adoption and maintenance: the family environment, the social/physical environment, the rise to independence and independent functioning. First, the family environment is the starting point for the adoption of HLHs, as already shown. The level of autonomy is limited and/or partial and everything is conducted in the natural setting of the family. When participants talk about the family environment, they refer to the diversity of activities they have experienced, its support and its role as a model. The second pillar includes the social environment and the physical environment (as well as the combination of the two). It is important to mention that the impact of these environments may have occurred simultaneously with or following the family environment, as it was impossible to obtain a precise chronological track on when they appeared as facilitators [41]. The results show that the social environment, such as peers and different human models other than the family, as well as the physical environment, such as parks near one’s home, act as winning conditions for the adoption and maintenance of HLHs. Third and fourth came the evolution of the participants’ career path, an important pillar in itself, which gradually places the individual at the center of their decisions (third pillar), as previous study demonstrated [42]. Individuals then manage their HLHs with an autonomy marked by independence (fourth pillar). This independence occurs at specific moments in the lives of our participants, such as leaving the family nest or starting post-secondary studies. What we observed at this last stage is the triangular link between psychological health, physical health and intrinsic awareness of the benefits of a healthy and active lifestyle. This desire and awareness of the need to be psychologically and physically healthy emerges for some when they have experienced a prolonged break from an active lifestyle and consequences have followed (e.g., injury that prevented PAP). For some, the good that was attached to this LAP dissipated and, in contrast, they indicated the importance of moving for the body and, more importantly, for mental well-being. For others, the sheer independence and autonomy that comes with it has made them realize that they need to take action to maintain active HLHs.

### 4.2. Major Findings and Scientific Literature

Our results show that the adoption and maintenance of HLHs follow the rise of autonomy and that, in parallel, several environments act at different key moments of the life course, from the family, to the community and the social and physical infrastructures, to the self. For ease of understanding, we present an ideal life course, in four major milestones, that synthesizes and illustrates what participants said.

#### 4.2.1. Milestone 1: Family Environment as the First Pillar of Healthy Lifestyle Habit Adoption

First, the family plays the largest role in the adoption of HLHs in children. Initially, the parents offer a variety of activities so that the child has a wide range of experiences related to the PAP. At this point, their autonomy is rather limited, as choices are more imposed by the family. By being active themselves, parents establish a culture of sport in the family and act as positive role models for both nutrition and physical activity. Several authors have reached the same conclusions. This is the case of Bergeron and Reyburn [43], who mention that the interactions between the family and the young person can lead to a strong identity that facilitates the adoption of HLHs. The aspect of culture is also present in other qualitative studies, such as Dagkas and Stathi’s [44], who interviewed adolescents about the factors that influence their PAP in and out of school. Their findings are consistent with those of our participants; they mention that parental culture becomes a mirror of the child’s involvement in PAP and that their support is crucial. Active parents then serve as role models for their children, which facilitates their children’s PAP [45,46,47,48]. In addition, the literature shows that parents who move with their young children also reinforce an active lifestyle in their children [45,49,50]. However, this element was not specified in many of our participants’ responses. It would have been interesting to question them to see if their parenting model involved parents who engaged in PA with them. In addition to this culture, parental support and encouragement also seem to be favorable and important in our ideal pathway. This actually agrees with Sallis et al. [51] and Hesketh et al. [52], in that these two aspects greatly influence a child’s PA level.

#### 4.2.2. Milestone 2: Social and Physical Environment as Facilitators to HLH Maintenance

Following this beginning of autonomy, the social and physical environment becomes increasingly important. In the case of the social environment, the young person tends to gravitate towards physical activities that he or she can practice with friends, for example, team sports that act as an incentive, as a motivator and that develop that community spirit. Salvy et al. [53] go in the same direction, mentioning that, during adolescence, friends become more important than family by greatly influencing our behaviors. To this end, their study found that, if friends practiced a PAP, it motivated the youth to practice one in turn and increased the likelihood that they would actually do so. Duncan et al. [54] reported similar results, stating that youths are more active when they are with friends. Unlike many studies that report the impact of an individual as a trigger/facilitator in PAP, such as a health and physical education teacher [55,56], our results do not identify this factor as a winning condition for the adoption and maintenance of HLHs. Could it be that our participants, who were predominantly from a family background conducive to HLH adoption, were less sensitive to the impact of a teacher solely because they were already aware of HLHs?

In the case of the physical environment, it has a positive effect on HLHs if there are facilities or infrastructure close to where people live. Our participants mentioned sports fields, parks, bike paths, etc., which they could choose to use. Several studies have reported on the impact of the physical environment on youths’ PAP. Dagkas and Stathi [44] highlight the role of the neighborhood in which we live. Similar to our results, their participants mention that proximity to facilities influenced their PAP. Xu et al. [57] make the same point; the variety, quality and proximity of facilities are important to adolescents’ level of PAP outside of school. In addition, the better a park is maintained, the more people enjoy using it [43].

Our participants go even further, as they mention the combination of the social and physical environment. They refer to available infrastructures, close by and used by the community and their peers; therefore, such facilities are alive with organized and animated activities, such as those related to sports leagues, for example. This means that it is not enough to have physical facilities, but rather to make them lively and to create a buzz and a community spirit around them. In the same logic, the literature review by Gadais, Boulanger, et al. [58] demonstrates the benefits of a good transportation system to facilitate the accessibility of these places. Authors report the importance of design features, such as safety, street quality, lighting, etc., in fostering an active community and developing positive perceptions of active transportation among youths [59,60]. Furthermore, authors argue that the built environment has effects on youths’ PAL based on the perceived level of social support [61]. This is also supported by D’Angelo et al. [62], who establish, in their study, a positive association between adolescents’ PAP and the role of peers, reinforced if their neighborhood holds many resources to move.

#### 4.2.3. Milestone 3: The Individual and His or Her Conscious Choices, Influenced by the Past and Dictated by the Desire to Be Physically and Mentally Healthy

This study shows that, as young people age, they become independent from their families and, thus, autonomous. With the arrival of this independence, they face completely different lifestyles, e.g., studies, work, living in an apartment, etc. Our results suggest that their decisions are now conscious, because they know that their choices influence their general health, unlike when they were young, when everything was done naturally (e.g., *I used to spend my weekends with my parents playing sports, I used to go out and play with my friends, without asking myself questions*). Some of our participants experienced a drop in PAP when they became independent, which caused an important trigger on the importance of having VHS (having experienced the negative effects of these changes in behavior, while realizing that it is necessary to take action to maintain HLHs). This does not appear to be a unique case. According to Leriche and Walczak [63], the decline in PAP among new CEGEP (Post-secondary level in Quebec, also known as college, preceding university by two years and giving access to it) students in Quebec is caused by poor time management. This perception of a lack of time would also be on the rise as their college career progresses. Despite this, our ideal pathway shows clear signs of maintenance (e.g., *I have a routine to keep moving, I have signed up for a new sport with friends*) and demonstrates a desire among participants to maintain a healthy and active lifestyle. Leriche and Walczak [63] refer to conscious, planned and purposeful choices, as in Ajzen’s and Godin’s models [13,14] with their notion of intention.

Furthermore, the intrinsic awareness of the benefits of HLHs leads our participants to maintain HLHs for the positive repercussions they have on a physical level, but especially on a psychological level. Indeed, the place of psychological health in relation to PAP greatly dictates their lifestyle. To achieve this, they keep a balance between the different spheres of life (e.g., social, physical, mental) and above all, perform PAP capable of bringing them this state of well-being. Many of them perform PA outdoors, with peers, and seek to feel pleasure during the activity. This close relationship between these two dimensions of health has been noted by many models, which raise consistent links with our results. First, it is recognized that engaging in outdoor physical activity has promising effects on well-being, in addition to bringing pleasure and satisfaction [64,65]. Then, authors mention that PAP has a distancing power, develops a sense of efficacy and self-esteem and that the social context plays a role in the psychological impacts [66,67,68,69,70]. Wankel [71], on the other hand, places pleasure in PAP as positively associated with psychological well-being. Finally, the link between physical and psychological health is explained through biological and physiological mechanisms, such as the secretion of certain hormones that have a role in anxiety [72] and, more specifically, in the hormonal regulation that decreases the physiological reactivity of stress [73].

#### 4.2.4. Milestone 4: A Timeline Based on Autonomy and Environments in the Ideal HLH Pathway

Overall, our data lead us to see an evolution in terms of autonomy and environments, which suggests the “ideal pathway” that we propose. Initially, the individual is more conditioned by his or her family, with much diversity and support, but little choice. Then, with the possibility of choosing which activity to invest according to their preferences and their feeling of self-efficacy, they favor an increased and possibly intrinsic motivation, as suggested by Deci and Ryan’s theory of self-determination [16]. Indeed, this theory relates the importance of having autonomy to increase the individual’s intrinsic motivation. Having autonomy-supporting contexts allow the person to choose in which opportunities to invest and to obtain positive feedback regarding this choice promotes motivation. Then, the social and physical environment becomes very important. The diversity and proximity of infrastructure has a positive effect on healthy and active living. Even better, the data suggest that it is the social support related to the built environment that can make a difference.

Finally, the ideal pathway ends with the individual at the center of their decisions, in complete autonomy. At this stage, the individual’s extensive knowledge of HLHs helps him or her make choices, while relying on the well-being that the PAP brings. The individual is still constrained by their environment; hence, the importance of using certain maintenance mechanisms, such as routine, friends, pleasure, the outdoors and balance among the spheres of life.

### 4.3. Research Perspectives Based on the Results of the Study

Our results suggest that the winning conditions for the adoption and maintenance of HLHs vary according to a multitude of factors, following an evolution under several environments (e.g., family, social, physical and individual) that change over the course of the individual’s life. This leads us to ask several questions. What, of all these winning conditions, that is, of all the factors that have a positive impact on HLHs, plays the key role in long-term maintenance? Is it the accumulation of all these facilitating factors or is it the key role of some of them, arriving at the right time in a person’s life? Overall, we found many similarities between the two group profiles under study, which led us to believe that our results follow certain regularities. However, how can we know the degree of impact for these winning conditions according to the age of the individual? It would be interesting to investigate the different phases of development, from childhood to adulthood, to better understand the factors that influence healthy lifestyles in people’s lives. Studies in this area would allow us to better target the interventions to be carried out with the desired clientele in health promotion.

On another note, our methodology, despite its small number of participants, has demonstrated its ability to obtain encouraging results for the field. Continuing to work on and refining qualitative designs to obtain more information on the long-term aspect seems to be a promising avenue for this type of research study. The originality of our design, in its methodology and in the preliminary modelling it proposes (storytelling), could help renew approaches in health promotion. Conducting longitudinal quantitative studies is very costly in terms of time and money. For this reason, it becomes relevant to work on qualitative specifications that open the methodology on long-term maintenance, investigating the life courses of people to understand what has mattered to them and made a difference. To this end, opting for studies with other profiles than those discussed here, in order to observe similarities or disparities, could provide a more complete picture of our understanding of the subject and would allow us to attempt modelling adapted to various social contexts.

Finally, our participants’ view of their lifestyle suggests the important place that psychological health occupies in their lives. Therefore, this dimension of health should be taken into account in future research.

### 4.4. Limits

One of the limitations of this research is the size of our sample. Initially, four group profiles were selected, but, in the end, only two profiles were selected (two groups at the Bachelor’s level in PE and 1 group at the Master’s level), for a total of 25 participants. This decline occurred as a result of the coronavirus (COVID-19) pandemic, which restricted access to our participants for group interviews. A larger sample size, as well as the more diverse profile of participants (e.g., not necessarily in favor of PAP) would have brought richness to the results, incorporating other desired profile types. Unfortunately, this more varied data collection was not possible.

Another limitation is related to possible social biases among our participants, which may have encouraged their participation despite themselves. Indeed, although participation was not mandatory, the interviews were conducted at their university and the researcher was introduced by their professor, which may have encouraged some of them to participate. Another problem also arises with the responses themselves. The subject matter of the study, which asks to talk about one’s own past experiences, can be difficult, first to grasp, to remember and then to verbalize. Indeed, Brown et al. [74] point out that we find it difficult to evaluate what has made us change and learn. On the other hand, we believe that by being aware of the topics discussed because of their field of training, directly related to the topic under study, these participants are better equipped to reflect on them and to verbalize their ideas.

Finally, the last limitation is related to the difficulty of generalizing our results. Indeed, our sample of participants was purposive and was highly targeted in this study. It does not necessarily represent the entire population, although the consistency of our results with those of many authors suggests a strong potential for transferability. However, we believe that this aspect of our research is a strength in itself, as the objective was to raise the elements that contribute to the adoption and maintenance of HLHs, beyond the barriers recognized by the literature. It became more than relevant to investigate the backgrounds of people who had the potential to have a significant repertoire, experience and reflection in terms of HLHs, in order to better highlight the important determinants of the adoption and maintenance of HLHs.

## 5. Conclusions

This research, which was both inductive and exploratory in nature, sought to advance knowledge on what can lead to the adoption of HLHs and, more specifically, to their maintenance. The methodology allowed us to meet our research objectives, using a multi-step analysis, following the principles of Paillé and Mucchielli’s [37] conceptualizing categories method. At the end, an ideal pathway in the form of a preliminary modeling of the evolutionary mechanisms of HLH adoption and maintenance allowed us to answer our research question. This model suggests that a multitude of factors act on the individual and his or her lifestyle, such as several environmental factors (family, social, physical and individual), which brings into play several key phenomena in the adoption and maintenance of HLHs (e.g., physical and psychological health, peers, the outdoors, motivation, pleasure, intrinsic awareness of benefits) and this unfolds over the course of an increasing autonomy in the paths of our participants.

These results may be an interesting contribution to research, considering the lack of long-term evidence in the field of health promotion. The impact of the different winning conditions on adopting and maintaining HLHs, conditions put forward in this research study, remains to be further investigated. One of the ways that seems promising to achieve this is to study them according to the phases of development, from childhood to adulthood. We believe that the present study is part of an evolution of methodological practices in the field of health promotion, with a design that tends to raise questions about the sustainability of HLHs. Therefore, we hope that this research study helps to inspire the methodology of future studies. Indeed, the conclusions drawn from this project could guide physical educators towards teaching strategies aimed at the diversity of the proposed activities, pleasure, the role of peers, motivation, etc. In addition, the creation of partnerships between the school and parents could be an interesting course of action, especially since our results show the importance of the family environment in the adoption of HLHs. In the end, we hope that this study allows the emergence of qualitative or mixed research methods in order to open the horizons in our approach to the notion of adoption and maintenance of HLHs.

## Figures and Tables

**Figure 1 ijerph-18-11057-f001:**
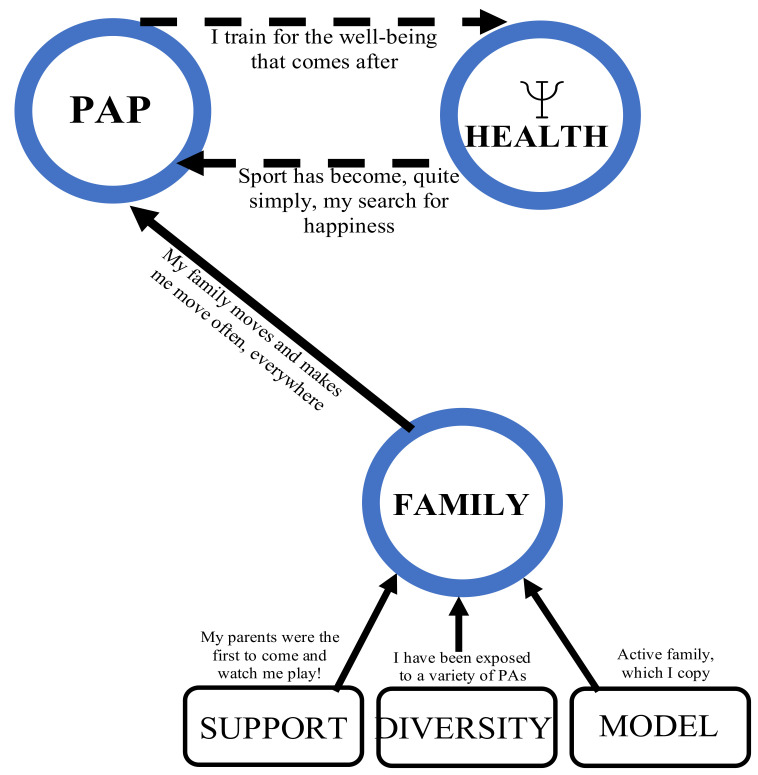
The three most important winning conditions for the adoption and maintenance of healthy lifestyle habits among participants in both profiles (the symbol below health in Figure 1 means Physical activity practice, family and mental health, the same as below).

**Figure 2 ijerph-18-11057-f002:**
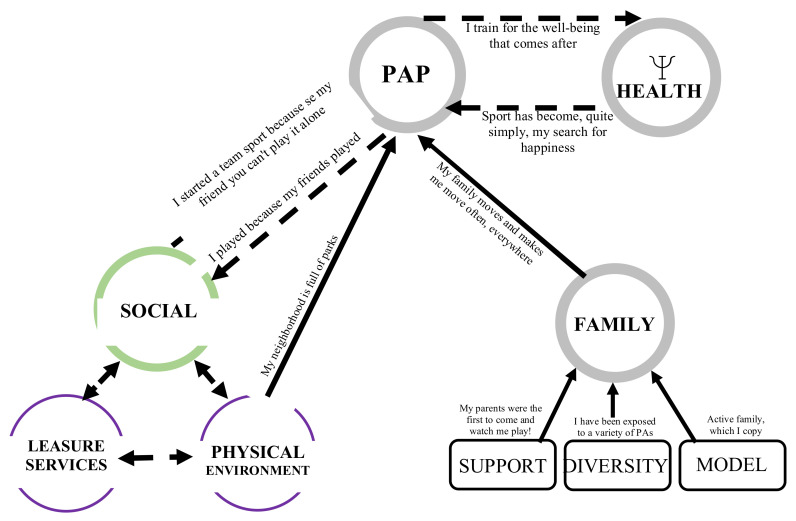
The social environment as a winning condition for the adoption and maintenance of healthy lifestyle habits.

**Figure 3 ijerph-18-11057-f003:**
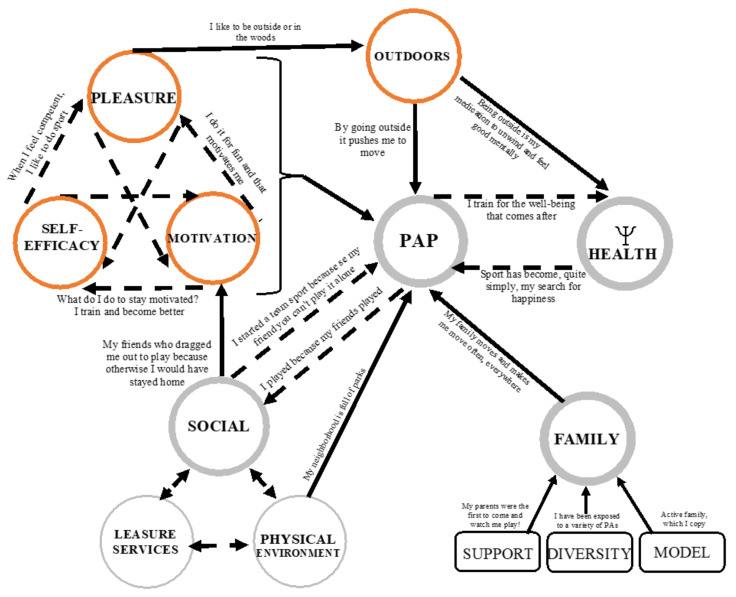
Role of enjoyment, self-efficacy, motivation and the outdoors in the adoption and maintenance of healthy lifestyle habits.

**Figure 4 ijerph-18-11057-f004:**
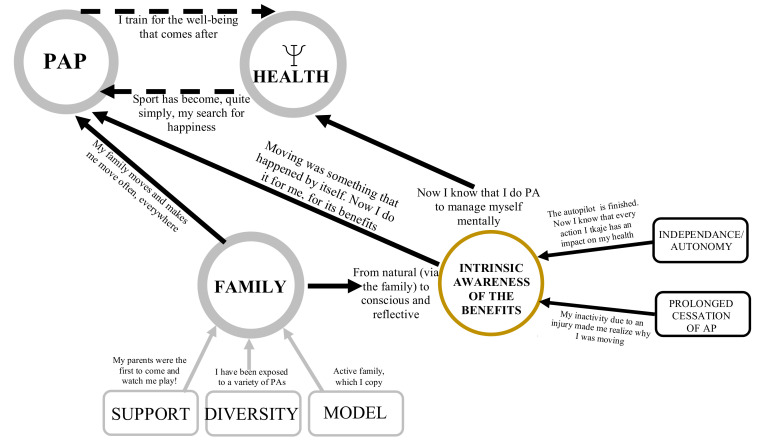
Intrinsic awareness of the benefits of physical activity as a driver for adoption and maintenance of healthy lifestyle habits.

**Figure 5 ijerph-18-11057-f005:**
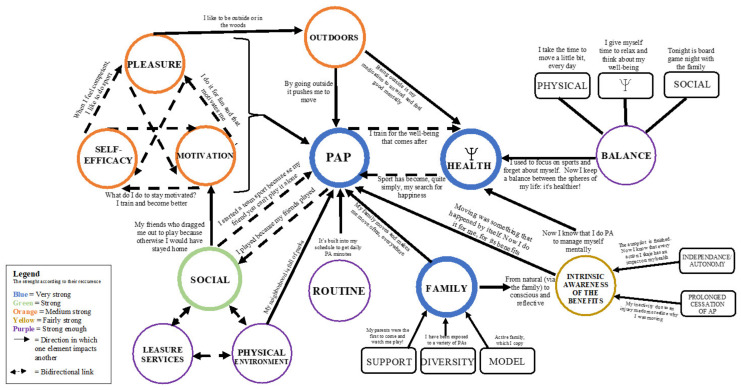
Summary of the winning conditions for the adoption and maintenance of healthy lifestyle habits.

**Figure 6 ijerph-18-11057-f006:**
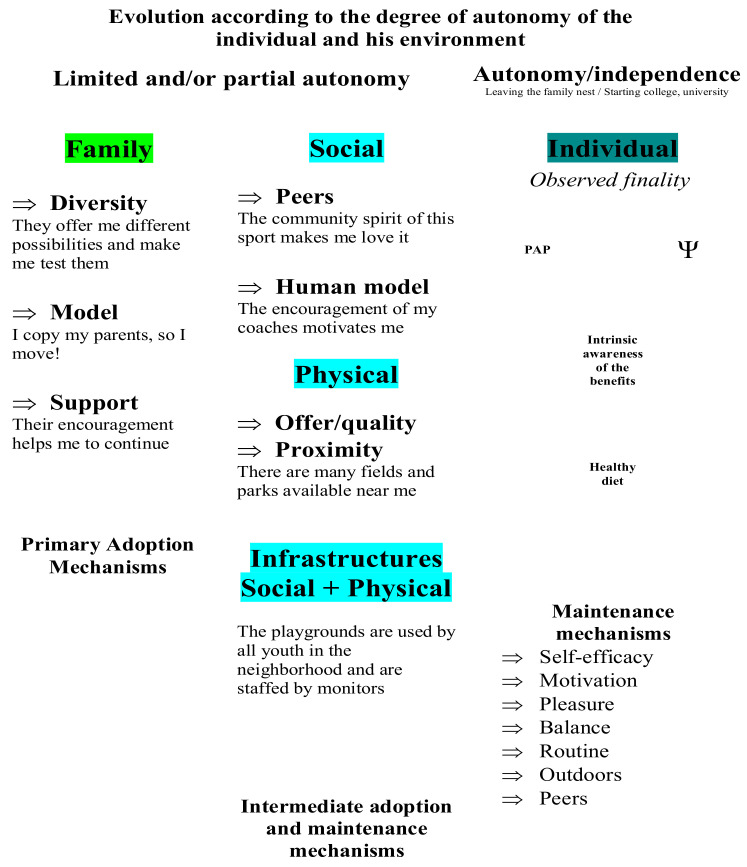
Conceptual model of evolutionary mechanisms of healthy lifestyle habit adoption and maintenance.

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
