# Peer review of "Winning Conditions for the Adoption and Maintenance of Long-Term Healthy Lifestyles According to Physical Activity Students"

_ijerph, 2021, doi:10.3390/ijerph182111057_

Round 1

Reviewer 1 Report

Comments and Suggestions for Authors

The authors have developed an investigation in relation to the influence of the winning conditions for the adoption and maintenance of long-term healthy lifestyles what I have found as very interesting to read.

First of all, I would like to congratulate the Authors the idea and the importance of the subject because in fact the literature has not yet documented the relative elements of the adoption of healthy lifestyle habits over the long term.

However, some aspects that improve the quality of the manuscript should be considered.

The introduction is considered adequate to the topic and with adequate references, as well as the approach to the objective of the study, but in my opinion is too long.

Regarding the material and method, the need to better describe the questions which the students gave the answers the person who interviewed them. For sure the size of the sample (25) it was limitation of the study, but pandemic destroys many projects, it is understandable.

Regarding the results: I have found very interesting the presentation of the figures and their development step by step. Regarding the Table 1 maybe authors will consider to present it in a different way or remove it. In my opinion the table is too large, and the information it contains are repeated in the text and figures anyway. The table takes a lot of space in the publication, which is very long, maybe The Authors will consider it as appendix.

Although the discussion is well structured in my opinion is too long.

The Authors should consider in the future to include in the research a few more group people, and maybe not only students who study particular profiles.

Congratulations again on the research, as it can provide relevant information in the field.

Author Response

RESPONSES TO REVIEWERS

Thanks for your kind comments. You perfectly understand what is the aim of this study and what we intend to do: study long term healthy lifestyles to complete and precise quantitative studies, beginning by good examples of healthy lifestyle professionals.

Following your comment, we tried to short a bit the introduction and keep the essential.

We provided a supplementary file to the manuscript with the list of questions we used into the interviews. A short version is available inside the manuscript L.229-231.

We follow your comment and remove table 1 from the text, however, we keep Appendix A and B to make the reference inside the text.

Following your comment, we tried to short a bit the discussion and keep the essential.

This comment is very relevant and we are totally aware of it. However, as a first study on this subject and with this methods, we decided to use more accessible participants for the study (students) with specific profile (positive to healthy lifestyles because professional of this area).

Your point is still pertinent and we did a reference to it into the limits L.608-612 (Initially, four group profiles were selected, but in the end, only two profiles were selected (two groups at the bachelor's level in PE and 1 group at the master's level), for a total of 25 participants. This decline occurred as a result of the coronavirus (COVID-19) pandemic, which restricted access to our participants for group interviews. A larger sample size would have brought richness to the results, incorporating other desired profile types. Unfortunately, this more varied data collection was not possible.)

Thanks for your encouraging comments! Yes

Reviewer 2 Report

I am glad I had a chance of reviewing this article. This is an interesting study and worth promoting. Undearneath I provided some comments and suggestions how to improve and broaden the context of the paper to reach for a bigger circulation of readers. 

Title is a little misleading - I would suggest changing 'winning' for something more suitable - maybe rephrase it using something like 'optimal"? or best mediating factors? But that is just my suggestion. 

Abstract is informative, but I had a feeling that the first part of the abstract was a bit lenghty. Maybe cut-off one or two sentences from the beginning.

Key words: I am not sure with putting there 'qualitative' because it does not say much what the authors had on their mind - qualitative analysis? qualitative PE? qualitative data? Maybe you should skip it.

The study is acctually quite interesting, despite the qualitative approach, which might turn to be its strenght. There are some limitations though, which the authors have been aware themeselves and which have been indicated by the authors in the final parts of the text. This is good, although I found this part a little too lenghty. 

Introduction is thorought and gives broad perspective on the theoretical frameworks and provides good backgroud for the further understanding of the research that the authors conducted. It is well-referenced, although with some opinions of the authors I would argue - for example 

 like on page 4. line 161-162 Added to this are the numerous limitations (e.g., lack of long-term evidence, need to complete research with mixed or qualitative designs) 

or later line 173 "Despite health promotion efforts, authors note a lack of ability and maintenance in health promotion intervention outcomes.." 

there have been many studies and papers on that issues published :

like "Tracking of PA from childhood to adulthood: a review" or "Tracking of Physical Activity from Early Childhood through Youth into Adulthood' or concerning youth sport experience "Paricipation in Organized youth sport as a predictor of adults physical activity: a 21-year longitudianal study"

These can also be used in Discussion section for broadening the context of discussing your own findings, as well as looking at the problem from a children's perspective "Do childern's expectations about future physical activity predict their physical activity in adulthood?" or "Will they stay fit and healthy? A three-year follow-up evaluation of a physical activity and health intervention in youth". 

In the Methods section research design, participants and recruiting process have been described well. Also reasearch methodology description is sufficient for someone who would like to duplicate the study protocol. Indeed sample size is small, but that is not by the foult of the authors. 

Results section is very long, and there is a lot of information. Perhaps there is no other way of presenting it, but at some point a reader looses track of all these findings. Also, something that raises my concern is the word "strenght' used by the authors in various moments of that part of the article in the figures (diagrams and grids) for example figure 5 - in the legends and in Appendix - how was this strenght assessed? 

Discussion part is well-written, though for the first few subsections it is presentation of authors own findings again and only from point 4.2 authors start discussing their results with the ones of the other authors. And here again, in some parts this discussion could be strenghten by using references from intervention studies in the family context for example "the analysis of social support level in foster families in the context of their leisure time activities' as parents'  support plays a crucial role in the development of life-long PA habits. This has been acknowledged by the authors, but later this role in supporting and encouraging young people to leisure PA rests heavily on PE teacher's support both in disabled children "Perceived facilitators and barriers for participation in leisure activities in children with disabilities: Perspectives of children, parents and professionals" and the other ones "Associations between adolescent's physical activity behaviour and their perceptions of parental, peer and teacher support". This could be brought into the light of the Discussion. 

I think enhancing the text and broadening the context of discussion of your findings will draw attention of broader scope of readers.   

Author Response

Please find attach in CC our responses to your comments.

Reviewer 3 Report

Promotion of health benefits is an important factor in the prevention and treatment of various diseases. Mastering the winning conditions is an essential part of successful management, but I think the scientific value of the study is low.

  1. Change in introduction is needed, introduction does not start of study focus.
  2. In general, the readability of the introduction is very low.
  3. There is a lot of information in this study, I think the authors should invite another scientist to the study with more experience in writing the manuscript.
  4. Generally, the structure of the manuscript is unusual.
  5. Where is the baseline characteristic of the participants?
  6. Why does the discussion have so many sections? It is uncommon.
  7. The conclusion must be direct and arising from the study.
  8. Less auto citations is needed

Author Response

Hi, 

Please find attached our responses to your comments.

Round 2

Reviewer 3 Report

I think the changes are enough